# FCHSD2 controls oncogenic ERK1/2 signaling outcome by regulating endocytic trafficking

**Guan-Yu Xiao**, **Sandra L. Schmid** *

Department of Cell Biology, University of Texas Southwestern Medical Center, Dallas, Texas, United States of America

* Sandra.Schmid@UTSouthwestern.edu

**Data Availability Statement:** All relevant data are within the paper and its Supporting Information files.

**Funding:** This work was supported by the National Institutes of Health (NIH) (R01 GM45455 and

## Abstract

The evolution of transformed cancer cells into metastatic tumors is, in part, driven by altered intracellular signaling downstream of receptor tyrosine kinases (RTKs). The surface levels and activity of RTKs are governed mainly through clathrin-mediated endocytosis (CME), endosomal recycling, or degradation. In turn, oncogenic signaling downstream of RTKs can reciprocally regulate endocytic trafficking by creating feedback loops in cells to enhance tumor progression. We previously showed that FCH/F-BAR and Double SH3 Domain-Containing Protein (FCHSD2) has a cancer-cell specific function in regulating CME in non-small-cell lung cancer (NSCLC) cells. Here, we report that FCHSD2 loss impacts recycling of the RTKs, epidermal growth factor receptor (EGFR) and proto-oncogene c-Met (MET), and shunts their trafficking into late endosomes and lysosomal degradation. Notably, FCHSD2 depletion results in the nuclear translocation of active extracellular signal-regulated kinase 1 and 2 (ERK1/2), leading to enhanced transcription and up-regulation of EGFR and MET. The small GTPase, Ras-related protein Rab-7A (Rab7), is essential for the FCHSD2 depletion-induced effects. Correspondingly, FCHSD2 loss correlates to higher tumor grades of NSCLC. Clinically, NSCLC patients expressing high FCHSD2 exhibit elevated survival, whereas patients with high Rab7 expression display decreased survival rates. Our study provides new insight into the molecular nexus for crosstalk between oncogenic signaling and RTK trafficking that controls cancer progression.

## Introduction

Non-small-cell lung cancer (NSCLC) is the leading cause of death among cancers worldwide [1]. Transformed NSCLC cells surreptitiously multiply while undergoing generations of selected evolution to acquire the characteristics of an aggressive and metastatic tumor. Increased metastatic activity is, in part, driven by altered signaling, often associated with activation of receptor tyrosine kinases (RTKs) [2–4]. The expression and activity of cell surface RTKs, in turn, is regulated predominantly through clathrin-mediated endocytosis (CME) [5–7], endosomal recycling, and/or degradation [8–10]. Therefore, a link between endocytic trafficking and cancer progression has been suggested [8,11,12]. Yet few studies have focused on cancer-cell–specific alterations in endocytic trafficking.

GM73165 to SLS). The funders had no role in study design, data collection and analysis, decision to publish, or preparation of the manuscript.

**Competing interests:** The authors have declared that no competing interests exist.

**Abbreviations:** AJCC, American Joint Committee on Cancer; BMP, bone morphogenetic protein; c-Fos, proto-oncogene c-Fos; c-Jun, proto-oncogene c-Jun; CME, clathrin-mediated endocytosis; EEA1, early endosome antigen 1; EGFR, epidermal growth factor receptor; ERK1/2, extracellular signal-regulated kinase 1 and 2; ETS1, protein C-ets-1; FCHSD2, FCH/F-BAR and Double SH3 Domain-Containing Protein; GAP, GTPase activating protein; GDP, guanosine diphosphate; GEF, guanine nucleotide exchange factor; HGF, hepatocyte growth factor; HRP, horseradish peroxidase; KD, knockdown; KRas, Kirsten Ras; LAMP1, lysosome-associated membrane glycoprotein 1; MAPK, mitogen-activated protein kinase; MEK1/2, dual specificity mitogen-activated protein kinase kinase 1 and 2; MET, proto-oncogene c-Met; MP1, MEK-binding partner 1; NSCLC, non-small-cell lung cancer; p-EGFR, phosphorylated EGFR; PFA, paraformaldehyde; qRT-PCR, quantitative reverse transcription PCR; Rab11, Ras-related protein Rab-11A; Rab7, Ras-related protein Rab-7A; RTK, receptor tyrosine kinase; shRNA, small hairpin RNA; siRNA, small interfering RNA; SOS, son of sevenless; STAT3, signal transducer and activator of transcription 3; TCEP, Tris (2-carboxyethyl) phosphine; TfnR, transferrin receptor.

CME is the major endocytic pathway that determines the rates of internalization of plasma membrane receptors, regulates their expression on the cell surface, and controls their downstream signaling activities [5–7]. We previously discovered that oncogenic signaling downstream of surface RTKs can, in turn, regulate CME and early recycling pathways by creating feedback loops that influence signaling, migration, and metastasis in NSCLC cells [13–15]. Moreover, some of these mechanisms for reciprocal crosstalk between signaling and the endocytic trafficking pathway appear to be specific for—or co-opted by—cancer cells to enhance tumor progression [14,15]. We have termed these cancer-specific changes in the endocytic machinery 'adaptive' endocytic trafficking and hypothesize that these 'gain-of-function' changes in endocytic trafficking contribute to cancer progression and metastasis [14].

The mechanisms that control the crosstalk between cargo (especially signaling receptors) and the endocytic machinery and their roles in cancer have not been explored. We recently discovered that the cancer-specific activation of FCH/F-BAR and Double SH3 Domain-Containing Protein (FCHSD2) downstream of extracellular signal-regulated kinases 1 and 2 (ERK1/2) contributes to adaptive CME in NSCLC cells [15], in this case by suppressing epidermal growth factor receptor (EGFR) signaling. Interestingly, its *Drosophila* ortholog, Nervous Wreck (*Nwk*), suppresses bone morphogenetic protein (BMP) signaling, but by regulating endosomal recycling at the synapse [16]. Whether human FCHSD2 also functions in endosomal trafficking of RTKs to regulate their oncogenic signaling from endosomes and whether this affects human tumor progression have not been studied.

To address this issue, we used HCC4017 and H1975 NSCLC cells, which exhibit oncogenic signaling pathways downstream of Kirsten Ras (KRas$^{G12C}$) or EGFR$^{T790M/L858R}$ mutations, respectively. These NCSLC cells were chosen due to their constitutive activation of the mitogen-activated protein kinase (MAPK) signaling pathway. We measured the effects of FCHSD2 depletion on the endocytic recycling and trafficking of the RTKs, EGFR and proto-oncogene c-Met (MET), and the consequences of these alterations on downstream signals. We demonstrated that FCHSD2 functions as a switch to regulate the trafficking pathway and destination of the RTKs through negative regulation of the small GTPase, Ras-related protein Rab-7A (Rab7). FCHSD2-dependent RTK trafficking controls the nuclear translocation of ERK1/2 signaling and expression of the RTKs. However, unlike its role in regulating CME [15], the FCHSD2 function in endosomal sorting does not require activation by ERK. Our study provides a novel mechanism of action by which protein traffic between endosomal compartments controls the outcome of ERK1/2 signaling and affects NSCLC progression.

## Results

### FCHSD2 regulates endosomal trafficking of transferrin receptor and EGFR in NSCLC cells

To test whether FCHSD2, like its *Drosophila* homologue, also functions in endosomal trafficking, we first assessed recycling of transferrin receptor (TfnR), a canonical marker for the quantification of endosomal trafficking [17]. To further determine which step(s) are affected, we measured TfnR recycling directly from early endosomes, following a 10-min pulse of internalized ligand or through perinuclear recycling endosomes following a 30-min pulse [18]. FCHSD2 knockdown (KD) by small interfering RNA (siRNA) selectively inhibited the extent of recycling of a 30-min pulse of TfnR, presumably from recycling endosomes, without affecting recycling directly from early endosomes (Fig 1A and 1B).

The effects of FCHSD2 on CME require its activation by ERK phosphorylation and appear to be specific to cancer cells [15]. Therefore, we tested the effect of ERK inhibitors on TfnR recycling in the presence and absence of FCHSD2, as well as the effect of FCHSD2 KD on

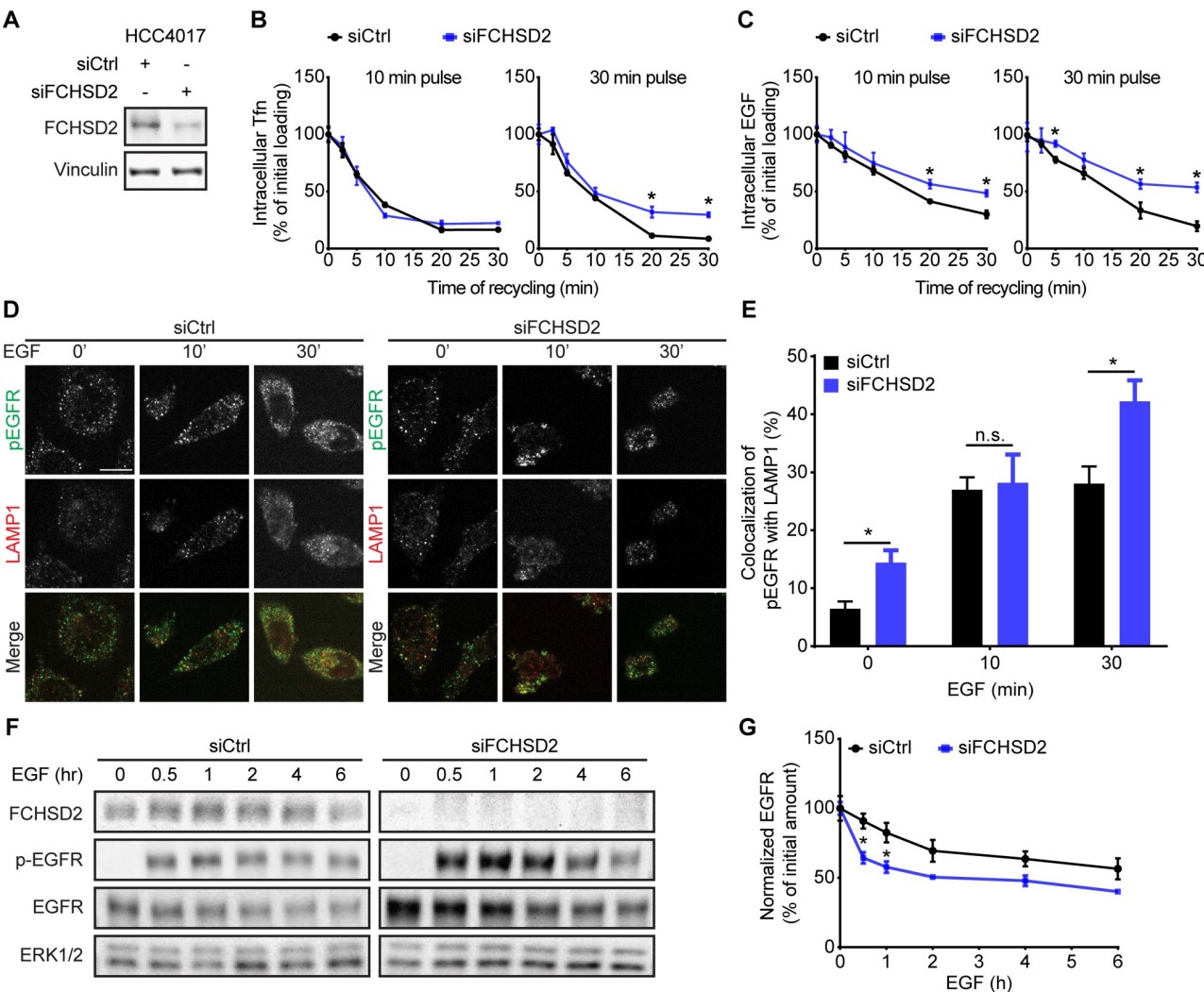

**Fig 1. FCHSD2 regulates TfnR and EGFR endocytic trafficking in NSCLC cells.** (A) The KD of FCHSD2 in control or FCHSD2 siRNA-treated HCC4017 cells. (B) and (C) Endocytic recycling of TfnR (B) or EGFR (C) was measured in control or FCHSD2 siRNA-treated HCC4017 cells. Cells were pulsed for 10 min or 30 min with 10 μg/ml biotinylated Tfn (B) or 20 ng/ml biotinylated EGF (C), stripped, and re-incubated at 37 °C for the indicated times before measuring the remaining intracellular Tfn or EGF. Percentage of recycled biotinylated Tfn or EGF was calculated relative to the initial loading. Data represent mean ± SEM (*n* = 3). Two-tailed Student *t* tests were used to assess statistical significance versus siCtrl. *$P < 0.05$, **$P < 0.005$, ***$P < 0.0005$. (D) Representative confocal images of pEGFR and LAMP1 immunofluorescence staining in control or FCHSD2 siRNA-treated HCC4017 cells. Cells were incubated with 20 ng/ml EGF for 30 min at 4 °C, washed, and re-incubated at 37 °C for the indicated times. Scale bar, 12.5 μm. (E) Colocalization of pEGFR and LAMP1 immunofluorescence staining in the cells as described in (D). Data were obtained from at least 40 cells in total/condition and represent mean ± SEM. Two-tailed Student *t* tests were used to assess statistical significance. *$P < 0.05$. (F) HCC4017 control or FCHSD2 siRNA-treated cells were stimulated with EGF (100 ng/ml) in the presence of cycloheximide (40 μg/ml) and incubated for the indicated times at 37 °C. (G) Quantification of EGFR/ERK intensity ratios in the cells as described in (F). Percentage of degraded EGFR was calculated relative to the initial amount. Data represent mean ± SEM (*n* = 3). Two-tailed Student *t* tests were used to assess statistical significance. *$P < 0.05$. The underlying data for this figure can be found in S1 Data. EGF, epidermal growth factor; EGFR, epidermal growth factor receptor; ERK, extracellular signal-regulated kinase 1 and 2; FCHSD2, FCH/F-BAR and Double SH3 Domain-Containing Protein; KD, knockdown; n.s., not significant; LAMP1, lysosome-associated membrane glycoprotein 1; NSCLC, non-small-cell lung cancer; siCtrl, control siRNA; siRNA, small interfering RNA; Tfn, transferrin; TfnR, transferrin receptor.

recycling in non-transformed ARPE-19 cells. Treatment of control siRNA with the ERK1/2 kinase inhibitor (SCH772984) resulted in a slight increase in the extent of TfnR recycling in both HCC4017 NSCLC cells and normal ARPE-19 cells (S1 Fig). As in cancer cells, siRNA-mediated KD of FCHSD2 significantly reduced TfnR recycling in ARPE-19 cells, although the

effects were somewhat smaller. ERK inhibition in siFCHSD2-treated cells increased the extents of TfnR recycling in both HCC4017 and ARPE-19 cells (S1 Fig), suggesting that its effects are due to other ERK substrate(s). These data suggest that, unlike its role in CME, the function of FCHSD2 in endosomal recycling was neither dependent on ERK1/2 activity nor cancer-cell–specific.

FCHSD2 KD also reduced the extent of EGFR recycling in both HCC4017 (Fig 1C) and H1975 (S2A and S2B Fig) NSCLC cells. However, unlike TfnR, EGFR is both recycled and degraded after internalization [19]. Therefore, to further explore which step(s) along the endocytic pathway were disrupted, we utilized an antibody specifically targeting phosphorylated EGFR (p-EGFR) at tyrosine 1068 to follow p-EGFR trafficking using immunofluorescence. We detected the accumulation of active EGFR in lysosome-associated membrane glycoprotein 1 (LAMP1)-positive late endosome/lysosomes upon FCHSD2 KD (Fig 1D and 1E, S2C and S2D Fig). In accordance with these results, EGF-stimulated EGFR degradation was enhanced upon FCHSD2 depletion (Fig 1F and 1G). These findings reveal additional roles for FCHSD2 in endosomal trafficking and sorting to lysosomes.

## FCHSD2 directs the endocytic trafficking of MET in NSCLC cells

We have shown that FCHSD2 regulates endocytosis of TfnR and EGFR and that it negatively regulates EGFR signaling from the cell surface [15]. Less studied but also highly associated with NSCLC progression is the RTK MET and its ligand hepatocyte growth factor (HGF). Moreover, it has been established that MET signaling requires endocytosis [20,21] and that it differentially signals from early versus late endosomes [22]. Given the role of FCHSD2 in endosomal trafficking, we speculate that FCHSD2 also functions in regulating the endocytic trafficking of MET. To test this hypothesis, we performed immunofluorescence to measure MET trafficking after HGF stimulation, using antibodies against MET, as well as different endosomal proteins (i.e., early endosome antigen 1 [EEA1], Ras-related protein Rab-11A [Rab11], and LAMP1, which mark, respectively, early endosomes, recycling endosomes, and late endosomes/lysosomes) (S3 and S4 Figs).

Loss of FCHSD2 resulted in the accumulation of MET in EEA1-positive early and LAMP1-positive late endosomes, with a corresponding decrease in colocalization with Rab11-positive recycling endosomes (Fig 2A). Additionally, FCHSD2 KD increased MET degradation following HGF stimulation (Fig 2B and 2C). These data demonstrate that FCHSD2 also regulates trafficking and degradation of MET.

## FCHSD2 KD-induced up-regulation of RTKs is independent of their activities

Paradoxically, we also noted that, despite decreased recycling and enhanced degradation of the RTKs following FCHSD2 KD, the steady-state levels of EGFR and MET were higher in the FCHSD2-deficient cells (Figs 1F and 2B). Unexpectedly, FCHSD2 KD resulted in increased levels of both *EGFR* and *MET* mRNA (Fig 3A), consistent with the increased expression seen at the protein level.

According to previous studies, the transfer of active RTKs to perinuclear endosomes can trigger the juxtanuclear activation of a weak signal transducer and activator of transcription 3 (STAT3) signal that leads to the required threshold of phosphorylation for nuclear translocation [23,24]. In turn, accumulation of nuclear p-STAT3 promotes the transcription of *HGF* and *c-Fos* [25], leading to up-regulation of EGFR [26] and MET [27,28]. To test whether this signaling pathway accounted for our findings, we treated cells with an EGFR inhibitor (afatinib) or a MET inhibitor (crizotinib). However, neither inhibitor affected the up-regulation of

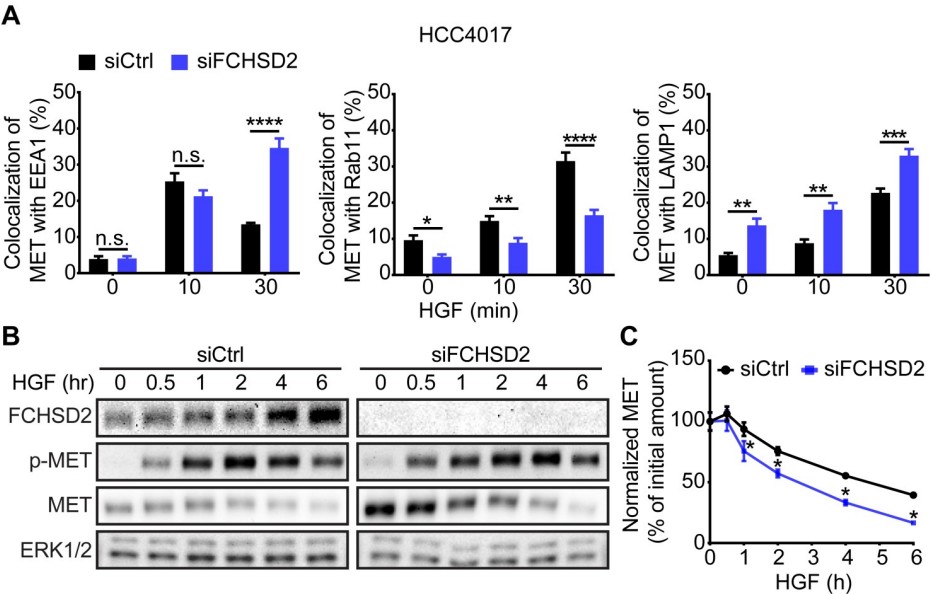

**Fig 2. FCHSD2 depletion alters the trafficking of MET.** (A) Quantification of the colocalization of MET with EEA1, Rab11, or LAMP1 immunofluorescence staining in control or FCHSD2 siRNA-treated HCC4017 cells. Cells were incubated with 1 μg/ml HGF for 30 min at 4 ˚C, washed, and re-incubated at 37 ˚C for the indicated times. Data were obtained from at least 40 cells in total/condition and represent mean ± SEM. Two-tailed Student $t$ tests were used to assess statistical significance. $^*P < 0.05$, $^{**}P < 0.005$, $^{***}P < 0.0005$, $^{****}P < 0.00005$. Representative confocal images are shown in S3 and S4 Figs. (B) HCC4017 control or FCHSD2 siRNA-treated cells were stimulated with HGF (100 ng/ml) in the presence of cycloheximide (40 μg/ml) and incubated at 37 ˚C for the indicated times. (C) Quantification of MET/ERK intensity ratios in the cells as described in (B). Percentage of degraded MET was calculated relative to the initial amount. Data represent mean ± SEM ($n$ = 3). Two-tailed Student $t$ tests were used to assess statistical significance. $^*P < 0.05$. The underlying data for this figure can be found in S1 Data. EEA1, early endosome antigen 1; ERK, extracellular signal-regulated kinase; FCHSD2, FCH/F-BAR and Double SH3 Domain-Containing Protein; HGF, hepatocyte growth factor; LAMP1, lysosome-associated membrane glycoprotein 1; MET, proto-oncogene c-Met; n.s., not significant; Rab11, Ras-related protein Rab-11A; siCtrl, control siRNA; siRNA, small interfering RNA.

RTKs in FCHSD2-depleted HCC4017 (Fig 3B and 3C) or H1975 (S5 Fig) NSCLC cells. Indeed, FCHSD2 KD decreased the level of p-STAT3 (Fig 3D) and was unable to trigger the transcription of STAT3 target genes, *HGF* and *c-Fos* (Fig 3E), perhaps as a result of its rerouting to the LAMP1-positive late endosomes/lysosomes at the expense of its accumulation in perinuclear Rab11-positive recycling endosomes (Fig 2A).

## ERK1/2 activity is responsible for the FCHSD2 KD-induced RTK up-regulation

Having ruled out STAT3 signaling and, indeed, the activities of the RTKs themselves, we next looked for alterations in the steady-state activity of other signaling pathways that might account for the up-regulation of EGFR and MET upon FCHSD2 KD. We found that FCHSD2 depletion specifically increased ERK1/2 but not Akt phosphorylation at steady state in HCC4017 cells (Fig 4A and 4B). We also observed that FCHSD2 deficiency in the HCC4017 cells stably expressing small hairpin RNAs (shRNAs) constitutively increased MET expression and ERK1/2 activity (S6 Fig). Notably, FCHSD2 KD significantly increased the expression of proto-oncogene c-Jun (c-Jun) and phosphorylated c-Jun (p-c-Jun), although the ratio of p-c-Jun/c-Jun was unaffected (Fig 4A–4C). Constitutive activation of ERK1/2 signaling induces *c-Jun* transcription and sustains c-Jun stability and activity [29]; correspondingly, FCHSD2 KD enhanced the transcription of *c-Jun* mRNA (Fig 4C). Furthermore, loss of FCHSD2 specifically

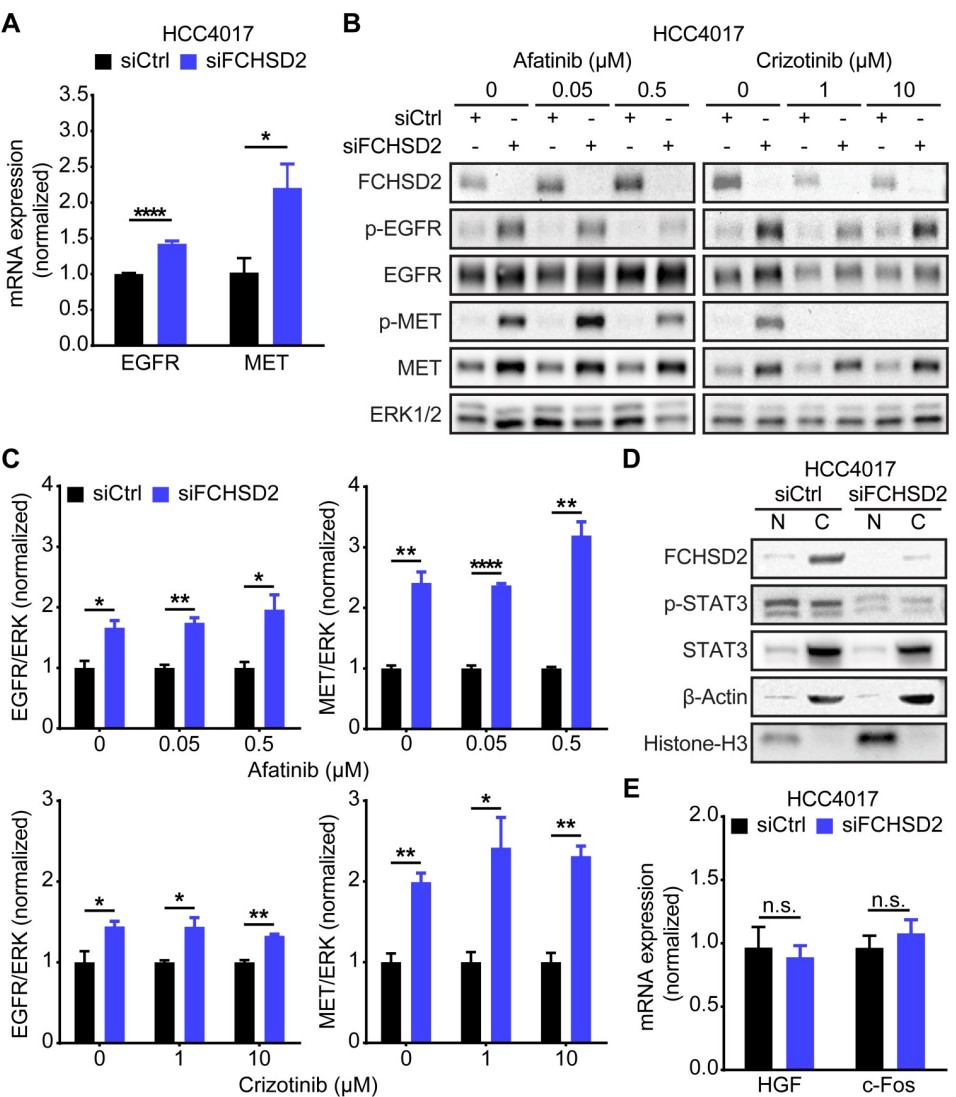

**Fig 3. FCHSD2 depletion-induced up-regulation of the RTKs is independent of their activities.** (A) FCHSD2 KD increases the transcription of *EGFR* and *MET* mRNA. All data were normalized to siCtrl and represent mean ± SEM ($n = 5$). Two-tailed Student $t$ tests were used to assess statistical significance. $^*P < 0.05$, $^{****}P < 0.00005$. (B) HCC4017 control or FCHSD2 siRNA-treated cells were incubated with EGFR inhibitor (afatinib) or MET inhibitor (crizotinib) at the indicated concentration for 24 h. Total ERK1/2 were served as the loading control as we have observed that the levels of total ERK1/2 remained consistent across different conditions (see Fig 4). (C) Quantification of EGFR/ERK or MET/ERK intensity ratios in the cells as described in (B). All data were normalized to siCtrl and represent mean ± SEM ($n = 3$). Two-tailed Student $t$ tests were used to assess statistical significance. $^*P < 0.05$, $^{**}P < 0.005$, $^{****}P < 0.00005$. (D) KD of FCHSD2 did not enhance translocation of phospho-STAT3 into the nucleus. Cell lysates from control or FCHSD2 siRNA-treated HCC4017 cells were subjected to fractionation. (E) Loss of FCHSD2 did not increase the transcription of phospho-STAT3 target genes, *HGF* and *c-Fos*. All data were normalized to siCtrl and represent mean ± SEM ($n = 5$). Two-tailed Student $t$ tests were used to assess statistical significance. The underlying data for this figure can be found in S1 Data. C, cytoplasmic fraction; c-Fos, proto-oncogene c-Fos; EGFR, epidermal growth factor receptor; ERK1/2, extracellular signal-regulated kinase 1 and 2; FCHSD2, FCH/F-BAR and Double SH3 Domain-Containing Protein; KD, knockdown; MET, proto-oncogene c-Met; N, nuclear fraction; n.s., not significant; RTK, receptor tyrosine kinase; siCtrl, control siRNA; siRNA, small interfering RNA; STAT3, signal transducer and activator of transcription 3.

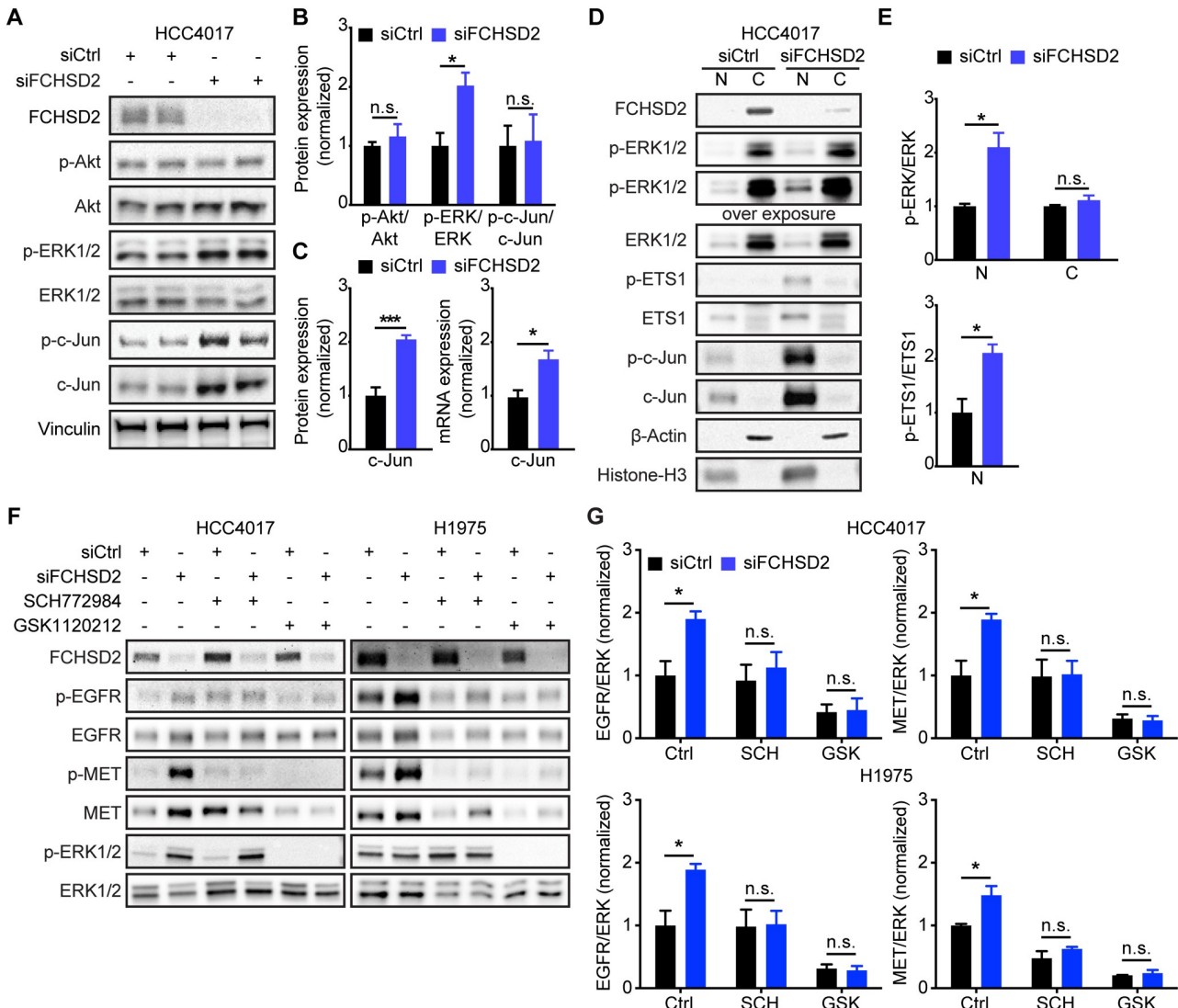

**Fig 4. ERK1/2 activity is essential for the RTK up-regulation induced by FCHSD2 depletion.** (A) Loss of FCHSD2 enhances ERK1/2 activity and c-Jun expression but not Akt activity in HCC4017 cells. (B) Quantification of signaling activities in the cells as described in (A). All data were normalized to siCtrl and represent mean ± SEM ($n$ = 6). Two-tailed Student $t$ tests were used to assess statistical significance. $^{*}P$ < 0.05. (C) KD of FCHSD2 increases c-Jun protein expression and mRNA transcription. The protein expression was determined by quantification of c-Jun/vinculin intensity ratios in the cells as described in (A). All data were normalized to siCtrl and represent mean ± SEM ($n$ = 5). Two-tailed Student $t$ tests were used to assess statistical significance. $^{*}P$ < 0.05, $^{***}P$ < 0.0005. (D) KD of FCHSD2 specifically enhances nuclear p-ERK1/2, p-ETS1, and c-Jun levels in HCC4017 cells. Cell lysates from control or FCHSD2 siRNA-treated HCC4017 cells were subjected to fractionation. (E) Quantification of p-ERK/ERK and p-ETS1/ETS1 intensity ratios in the cells as described in (D). All data were normalized to siCtrl and represent mean ± SEM ($n$ = 3). Two-tailed Student $t$ tests were used to assess statistical significance. $^{*}P$ < 0.05. (F) ERK or MEK inhibition disrupts the RTK up-regulation induced by FCHSD2 KD. ERK1/2 inhibitor (SCH772984, 1 μM) or MEK1/2 inhibitor (GSK1120212, 1 μM) was used to treat control or FCHSD2 siRNA-treated HCC4017 or H1975 cells for 72 h. (G) Quantification of EGFR/ERK or MET/ERK intensity ratios in the cells as described in (F). All data were normalized to siCtrl-Ctrl and represent mean ± SEM ($n$ = 3). Two-tailed Student $t$ tests were used to assess statistical significance. $^{*}P$ < 0.05. The underlying data for this figure can be found in S1 Data. C, cytoplasmic fraction; c-Jun, proto-oncogene c-Jun; Ctrl, vehicle control; EGFR, epidermal growth factor receptor; ERK1/2, extracellular signal-regulated kinase 1 and 2; ETS1, protein C-ets-1; FCHSD2, FCH/F-BAR and Double SH3 Domain-Containing Protein; GSK, GSK1120212; KD, knockdown; MEK, dual specificity mitogen-activated protein kinase kinase; MET, proto-oncogene c-Met; N, nuclear fraction; n.s., not significant; RTK, receptor tyrosine kinase; SCH, SCH772984; siCtrl, control siRNA; siRNA, small interfering RNA.

increased ERK1/2 activity in the nucleus, while the ratio of p-ERK/ERK in the cytoplasm remained unchanged (Fig 4D and 4E). In addition to enhancing c-Jun expression, the accumulation of nuclear p-ERK1/2 in FCHSD2-depleted cells promoted activity of the ERK1/2 target, protein C-ets-1 (ETS1) [30] (Fig 4D and 4E). Both c-Jun and ETS1 are known transcription factors for *EGFR* [26] and *MET* [28,31] and thus can contribute to the observed increase in transcription of *EGFR* and *MET* mRNA after FCHSD2 KD (Fig 3A).

To directly test whether ERK1/2 activity is required for increased expression of the RTKs in FCHSD2 KD cells, we used an ERK1/2 kinase inhibitor (SCH772984) and an inhibitor targeting the essential upstream kinase, dual specificity mitogen-activated protein kinase kinase 1 and 2 (MEK1/2) (GSK1120212). As predicted, both ERK1/2 and MEK1/2 inhibition disrupted the up-regulation of EGFR and MET in HCC4017 and H1975 cells (Fig 4F and 4G; note that inhibition of ERK activity does not correlate with inhibition of ERK phosphorylation by MEK). Together, these results suggest that increased ERK1/2 activity in the nucleus is essential for the effects of FCHSD2 depletion in NSCLC cells.

## Rab7 is required for the effects of FCHSD2 KD on RTK expression

Previous studies have shown that translocation of active ERK1/2 to the nucleus requires the recruitment of MEK1 to Rab7-positive endosomes, where MEK1 activates ERK1/2 signaling from late/perinuclear endosome compartments [32]. In addition, Rab7 supports endosome maturation and promotes endocytic trafficking toward late endosomes rather than recycling endosome compartments [33]. In agreement with previous research, Rab7 KD increased the rate of TfnR and EGFR recycling (Fig 5A). Strikingly, there was no difference between the effects of Rab7 KD alone and the depletion of both FCHSD2 and Rab7 on the TfnR and EGFR recycling (Fig 5A), indicating that the FCHSD2 KD-induced phenotype depends on the function of Rab7. Given that the activities of MEK1/2 and ERK1/2 are necessary for the effects of FCHSD2 KD, we further tested the consequences of Rab7 depletion on EGFR and MET expression. Importantly, Rab7 KD by 2 different siRNA pools abolished the up-regulation of EGFR and MET induced by FCHSD2 depletion (Fig 5B and S7 Fig).

The activity of the small GTPase, Rab7, is regulated by a switch between active GTP-bound (Rab7•GTP) and inactive GDP-bound (Rab7•GDP) states [33]. To assess the effect of FCHSD2 KD on Rab7 activity, we immunoprecipitated active Rab7 from HCC4017 cells using an antibody that specifically recognizes Rab7•GTP. We found significantly higher levels of active Rab7 in the FCHSD2-deficient cells (Fig 5C). These data suggest that FCHSD2 controls expression and trafficking of the RTKs by negatively regulating Rab7. Thus, FCHSD2 and Rab7 play antagonistic roles in regulating endosomal trafficking.

## FCHSD2 and Rab7 differentially affect lung cancer progression

Our studies have revealed a multifaceted function for FCHSD2 in the crosstalk between endocytic trafficking and oncogenic signaling in NSCLC cells (Fig 6A). We previously showed that FCHSD2 has a cancer-cell–specific function in positively regulating CME downstream of ERK1/2 activity. Here, we show that FCHSD2 plays a general role in regulating endosomal trafficking by negatively modulating Rab7 activity. Together, these activities establish FCHSD2 as a key regulator in the outcome of oncogenic ERK1/2 signaling by controlling the trafficking and expression of EGFR and MET. Given that FCHSD2 KD dramatically increased the proliferation and the migration activities of NSCLC cells [15], FCHSD2 may function as a negative regulator for human lung tumor growth. In contrast, Rab7 is thought to favor lung cancer progression. To investigate the correlation between FCHSD2 activity and lung tumor progression, we directly measured the protein expression level of FCHSD2 in tumor tissues from lung

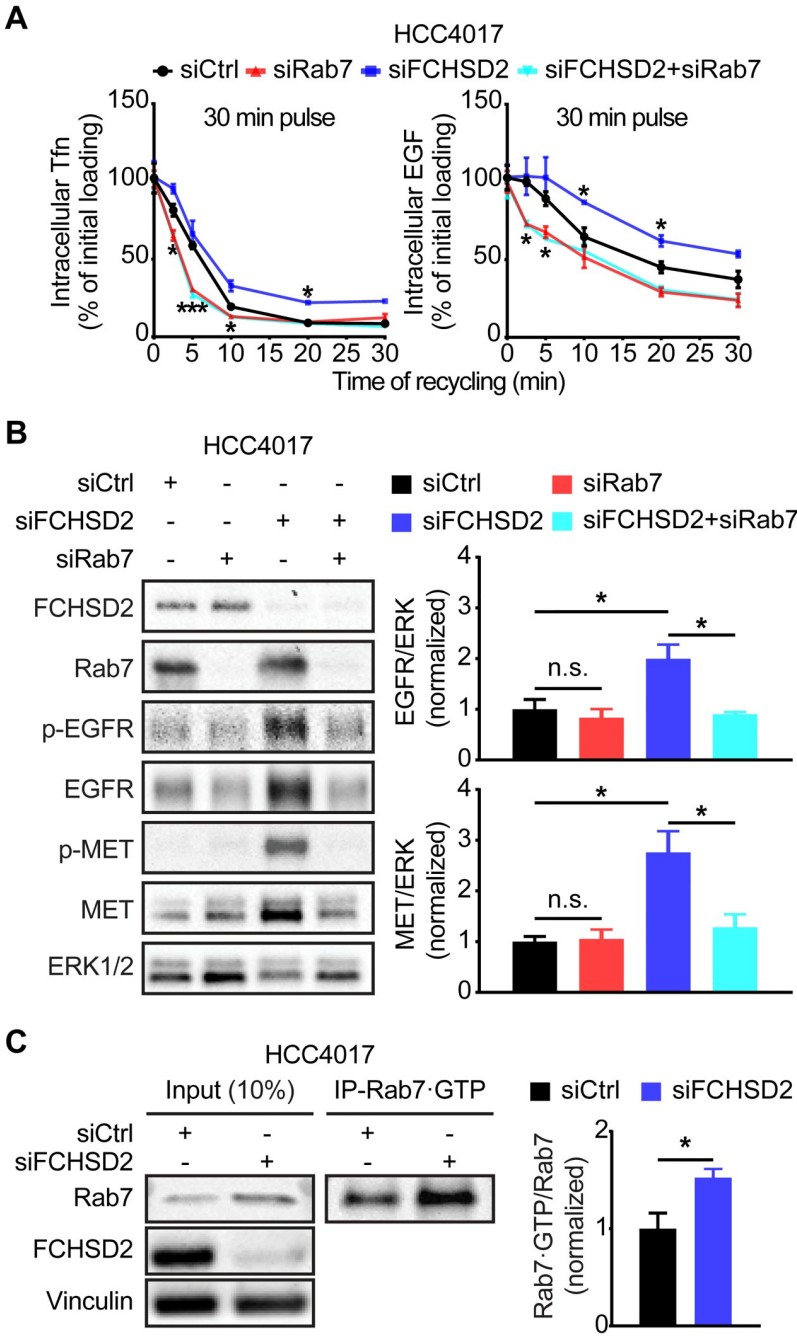

**Fig 5. Rab7 is required for FCHSD2 depletion-induced up-regulation of the RTKs.** (A) Endocytic recycling of TfnR or EGFR was measured in control, Rab7, FCHSD2, or both siRNA-treated HCC4017 cells. Cells were pulsed for 30 min with 10 μg/ml biotinylated Tfn or 20 ng/ml biotinylated EGF, stripped, and re-incubated at 37 ˚C for the indicated times before measuring the remaining intracellular Tfn or EGF. Percentage of recycled biotinylated Tfn or EGF was calculated relative to the initial loading. Data represent mean ± SEM (*n* = 3). Two-tailed Student *t* tests were used to assess statistical significance versus siCtrl. *P < 0.05, ***P < 0.0005. (B) Rab7 KD abolishes the RTK up-regulation induced by FCHSD2 depletion. Quantification of EGFR/ERK or MET/ERK intensity ratios in the cells was measured. All data were normalized to siCtrl and represent mean ± SEM (*n* = 3). Two-tailed Student *t* tests were used to assess statistical significance. *P < 0.05. (C) FCHSD2 depletion promotes the activity of Rab7. Cell lysates from control or FCHSD2 siRNA-treated HCC4017 cells were immunoprecipitated with anti-Rab7·GTP (active form of Rab7) antibody. The indicated proteins were detected. Quantification of Rab7·GTP/input Rab7 intensity ratios in the cells was measured. All data were normalized to siCtrl and represent mean ± SEM (*n* = 3). Two-tailed Student *t* tests were used to assess statistical significance. *P < 0.05. The underlying data for this figure can be found in S1 Data. EGF,

epidermal growth factor; EGFR, epidermal growth factor receptor; ERK, extracellular signal-regulated kinase;
FCHSD2, FCH/F-BAR and Double SH3 Domain-Containing Protein; KD, knockdown; MET, proto-oncogene c-Met;
n.s., not significant; Rab7, Ras-related protein Rab-7A; RTK, receptor tyrosine kinase; siCtrl, control siRNA; siRNA,
small interfering RNA; Tfn, transferrin; TfnR, transferrin receptor.

adenocarcinoma patients. As expected from our in vitro findings, FCHSD2 expression is gradually decreased in higher grades of lung adenocarcinoma tumors (Fig 6B). Further studies will be needed to compare FCHSD2 expression in normal versus adenocarcinoma cancerous lung tissues.

Finally, we examined the relationship between FCHSD2 and Rab7 expression and NSCLC patient survival by mining clinical data. The expression of these 2 factors were highly associated with lung cancer patient survival rates (Fig 6C). Patients with relatively high FCHSD2 expression had significantly better survival rates than those in the low-expression group (Fig 6C). In contrast, Rab7 expression had the opposite correlation with patient survival rates (Fig 6C). Notably, the correlation between the expression of FCHSD2 or Rab7 with survival rates was more prominent in lung adenocarcinoma patients (Fig 6C). Although they are correlative, these data are consistent with our in vitro findings that FCHSD2 functions as a negative regulator of Rab7 and controls lung cancer aggressiveness.

## Discussion

Endocytic trafficking regulates the expression and activity of RTKs and modulates their downstream signaling to maintain cell homeostasis [34]. We previously reported that activation of FCHSD2 by ERK1/2 phosphorylation increases the rate of TfnR and EGFR internalization by CME and suppresses signaling from cell surface EGFRs, specifically in cancer cells [15]. Here, we show that FCHSD2, like its *Drosophila* orthologue *Nwk* [16], also enhances the trafficking of internalized RTKs from the early endosomes toward recycling endosomes and reduces their trafficking to late endosomes/lysosomes. These activities are independent of ERK activation, are not specific to cancer cells, and involve the negative regulation of Rab7. Together, these FCHSD2-dependent changes enhance trafficking of RTKs through the early and recycling endocytic pathways to suppress signaling downstream of activated RTKs.

Overexpressed RTKs are a common feature among different types of cancers and widely considered favorable for tumor progression [35]. In particular, MET—the HGF receptor—is up-regulated in approximately 50% of NSCLC, particularly in lung adenocarcinomas (72.3%) [36]. Hyperactivity of MET and its dependent invasive growth signals are a general feature of highly aggressive tumors and are associated with poor survival [37]. Moreover, the activation of MET and its downstream signaling is dependent on trafficking through both early and late endosomes [21,38]. FCHSD2 depletion resulted in increased expression of MET at both the transcriptional and protein levels, as well as decreased recycling. Together, these effects resulted in the accumulation of MET in both early and late endosome/lysosome compartments, despite increased degradation.

Our findings show that the functions of Rab7 and FCHSD2 in endosomal trafficking are antagonistic. Rab7 is a ubiquitously expressed member of the Rab family of small GTPases that regulates the maturation of early endosomes into late endosomes, the fusion of late endosomes with lysosomes in the perinuclear region, and lysosomal biogenesis [33]. The effects of FCHSD2 KD on endosomal trafficking and consequent up-regulation of EGFR and MET expression are dependent on Rab7, and FCHSD2 appears to negatively regulate Rab7 activation. It remains to be determined whether this occurs via direct interactions between FCHSD2 and Rab7 or, more

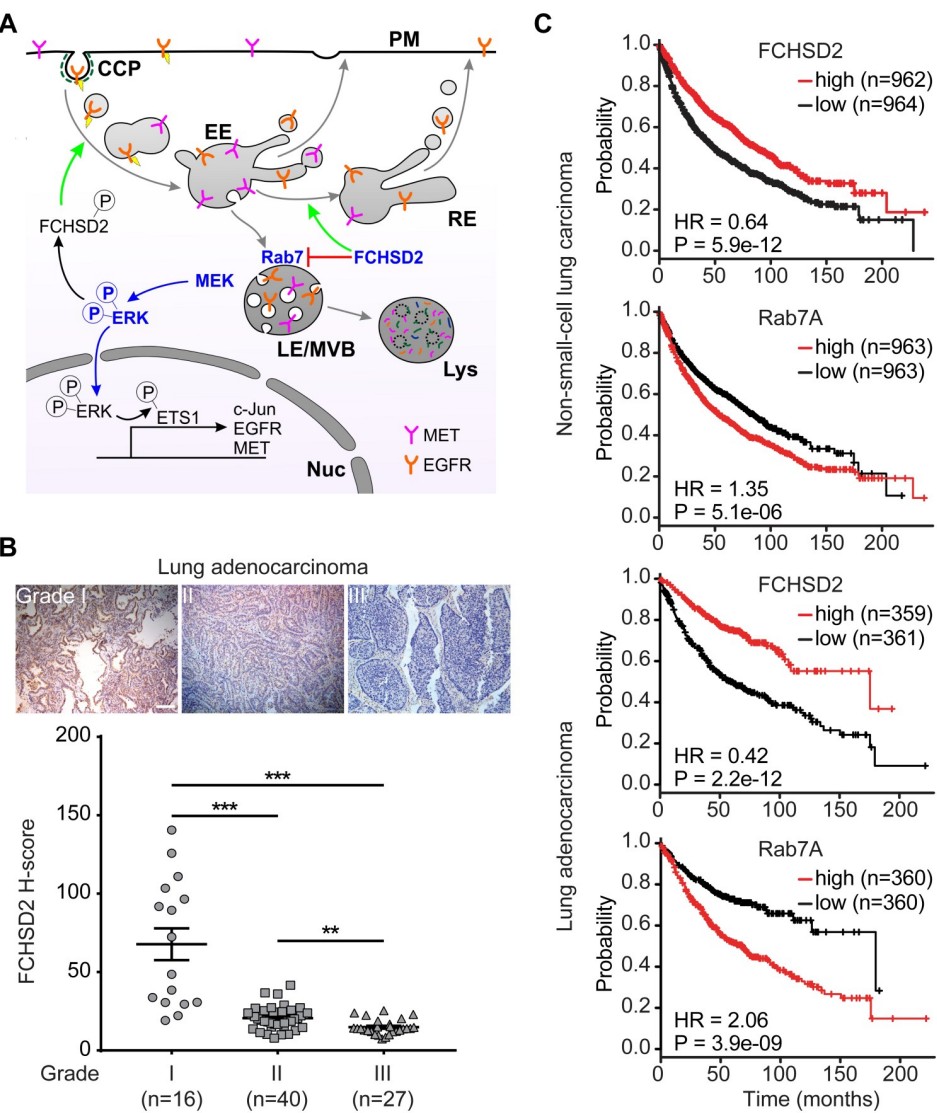

**Fig 6. FCHSD2 and Rab7 reciprocally regulate endocytic trafficking and lung cancer progression.** (A) FCHSD2 regulates multiple steps in endocytic trafficking. We previously showed that activation of FCHSD2 downstream of ERK1/2 increases the rate of clathrin-coated pit initiation and CME in NSCLC cells. Here, we report that FCHSD2 also increases the fraction of RTK trafficking from early endosomes to recycling endosomes and negatively regulates Rab7 activity, maturation of late endosomes/multivesicular bodies, and trafficking to lysosomes. Together, these activities of FCHSD2 increase the flux of RTKs through early endocytic pathways, thus altering their downstream signaling. Loss of FCHSD2 results in the accumulation of RTKs in late endosomes/lysosomes, increases levels of activated ERK1/2 in the nucleus, and enhances transcription and expression of c-Jun, EGFR, and MET. (B) Immunohistochemistry images and quantification (expressed as H-score) of FCHSD2 staining in representative lung tumor tissues. Scale bar, 100 μm. Two-tailed Student $t$ tests were used to assess statistical significance. $^{**}P < 0.005$, $^{***}P < 0.0005$. (C) Kaplan-Meier survival analysis of NSCLC or lung adenocarcinoma patients was performed in FCHSD2 or Rab7 high- and low-expression cohorts. The underlying data for this figure can be found in S1 Data. CCP, clathrin-coated pit; c-Jun, proto-oncogene c-Jun; CME, clathrin-mediated endocytosis; EE, early endosome; EGFR, epidermal growth factor receptor; ERK1/2, extracellular signal-regulated kinase 1 and 2; ETS1, protein C-ets-1; FCHSD2, FCH/F-BAR and Double SH3 Domain-Containing Protein; HR, hazard ratio; LE, late endosome; Lys, lysosomes; MET, proto-oncogene c-Met; MVB, multivesicular body; NSCLC, non-small-cell lung cancer; PM, plasma membrane; Rab7, Ras-related protein Rab-7A; RE, recycling endosome; RTK, receptor tyrosine kinase.

likely, indirectly through the negative regulation of Rab7 guanine nucleotide exchange factors (GEFs) or the positive regulation of Rab7 GTPase activating proteins (GAPs).

Our mechanistic studies of FCHSD2 and Rab7 function in NSCLC cell lines were consistent with clinical databases showing that high levels of FCHSD2 expression and, reciprocally, low levels of Rab7 expression correlate with improved survival rates. This was especially the case among lung adenocarcinoma patients, whose cancers are frequently driven by oncogenic mutations that activate MAPK signaling [39,40]. Correspondingly, we also observed a progressive decrease in FCHSD2 protein levels detected by immunohistochemistry in higher grade of lung adenocarcinomas.

Our studies were focused on 2 NSCLC cell lines, HCC4017 and H1975, that also exhibit activated MAPK signaling due to mutations in $KRas^{G12C}$ or $EGFR^{T790M/L858R}$, respectively. Whether and to what extent FCHSD2 expression and/or activation plays a role in regulating membrane trafficking and ERK1/2 signaling in other NSCLC cell lines or other cancer types are worth investigating in future experiments. Nevertheless, that these 2 proteins converge on regulating trafficking between early endosomes, recycling endosomes, and late endosomes suggests an important role for endosomal trafficking in regulating oncogenic signaling in cancer cells and tumor progression.

FCHSD2 KD led to increased steady-state expression levels of the oncogenes MET, EGFR, and c-Jun, as well as an increase in the steady-state activation of ERK1/2 specifically in the nucleus. Although previous studies have shown that internalization of active EGFR and MET is required for their signaling from endosomal compartments—which, in turn, triggers transcription and expression of the RTKs [23–27]—the increases in MET and EGFR expression were not dependent on either of their kinase activities. Both, however, required ERK1/2 activity, and we found that ETS1, a transcription factor known to regulate EGFR and MET expression downstream of ERK, is activated upon FCHSD2 KD. Further studies are needed to confirm that changes in EGFR and MET expression at the protein level, e.g., under conditions of ERK1/2 inactivation and Rab7 KD, correlate with changes at the transcriptional level.

A previous study showed that KRas can be constitutively internalized via CME and activated on Rab7-positive late endosomes [41]. There, the Rab7-endosome–associated adaptor protein p14 and the scaffolding protein MEK-binding partner 1 (MP1) tether MEK and ERK1/2 for downstream activation of ERK1/2 [42]. We speculate that, even if inactive, the high local concentrations of EGFR and MET that accumulate on late endosomes after FCHSD2 KD may be sufficient to recruit the GEF protein, son of sevenless (SOS) and activate KRas. Consistent with this, basal Ras activation has also been reported in Neimann-Pick C fibroblasts in which late endosomal trafficking is perturbed [43]. Interestingly, we noted that the magnitude of the effects of FCHSD2 KD were generally greater in the HCC4017 cells that express activated $KRas^{G12C}$ than in H1975 cells.

Collectively, our findings define an endocytic trafficking pathway regulated by FCHSD2 and Rab7 that functions to control RTK expression, oncogenic signal transduction, and NSCLC progression. Knowledge of these trafficking pathways and their (dys)regulation during cancer progression could help us to identify potential new therapeutic targets for the prevention of aggressive cancers and/or prognostic indicators that can guide lung cancer treatment.

## Materials and methods

### Cell culture and chemicals

HCC4017 ($KRas^{G12C}$, $EGFR^{WT}$) and H1975 ($KRas^{WT}$, $EGFR^{T790M/L858R}$) NSCLC cells (from John Minna, UT Southwestern Medical Center, Dallas, TX) were grown in RPMI 1640 (Thermo Fisher Scientific, Waltham, MA) supplemented with 10% (vol/vol) FBS. ARPE-19

cells (from ATCC, Manassas, VA) were cultivated in DMEM/F12 (Thermo Fisher Scientific) supplemented with 10% (vol/vol) FBS. The recombinant human EGF used in this study was from Thermo Fisher Scientific, Waltham, MA, and the recombinant human HGF was generously provided by Drs. Emiko Uchikawa and Xiaochen Bai (UT Southwestern Medical Center, Dallas, TX). The cycloheximide was from MilliporeSigma, Burlington, MA. The EGFR inhibitor afatinib, the MET inhibitor crizotinib, and the ERK inhibitor SCH772984 were from Selleck Chemicals, Houston, TX. The MEK inhibitor GSK1120212 was from MedChemExpress, Monmouth Junction, NJ.

## RNA interference and stable cell line generation

Cells were treated with the siRNA pool targeting FCHSD2 (#J-021240-10, #J-021240-11, #J-021240-12; Dharmacon, Lafayette, CO) or Rab7 (#LU-010388-00, Dharmacon or #SASI_Hs01_00104357, #SASI_Hs01_00104360, #SASI_Hs01_00104361, MilliporeSigma) using RNAiMAX (Thermo Fisher Scientific) to silence the endogenous protein. Briefly, 50 nM of the indicated siRNA pool and 6.5 μl of RNAiMAX reagent were added in 1 ml of OptiMEM (Thermo Fisher Scientific) to each well of a 6-well plate and incubated for 20 min at room temperature. Cells were resuspended in 1 ml of culture medium, seeded in each well of a 6-well plate at 20%–30% confluency containing the mixed siRNA–lipid complex, and incubated for 48–72 h, followed by experiments. The AllStars Negative siRNA non-targeting sequence was purchased from Qiagen, Hilden, Germany (#SI03650318).

shCtrl (#RHS4346; Dharmacon) or the pool of shFCHSD2 lentiviral constructs (#RHS4430-200190839, #RHS4430-200163596, #RHS4430-200278354; Dharmacon) transduction in HCC4017 or H1975 cells was performed using lentiviral vectors. The lentiviral vectors and lentiviral packing plasmids (pSPAX2 [Gag/Pol] and pMD2G) were co-transfected into HEK293T cells using Lipofectamine 2000 (Thermo Fisher Scientific) according to the manufacturer's instructions for virus production. The virus-conditioned media were used to infect target cells in the presence of 10 μg/ml polybrene (MilliporeSigma). After infection for 48 h, the cells were grown in media containing puromycin (2.5 μg/ml) for 2 wk before experiments.

## Western blotting and analyses

Cells cultured in each well of a 6-well plate at 80% confluency were washed 3 times with PBS and harvested/resuspended in 150–200 μl of 2× Laemmli buffer (Bio-Rad, Hercules, CA). The cell lysate was boiled for 10 min and loaded onto an SDS gel. After transferring to a nitrocellulose membrane (Bio-Rad), membranes were blocked with 5% milk in TBST buffer and were probed with antibodies diluted in 5% BSA in TBST buffer against the following proteins: FCHSD2 (#PA5-58432, 1:500; Thermo Fisher Scientific), Rab11 (#5589S, 1:1000; Cell Signaling, Danvers, MA), Rab7 (#9367S, 1:1000; Cell Signaling), p-EGFR Y1068 (#3777S, 1:1000; Cell Signaling), EGFR (#4267S, 1:1000; Cell Signaling), p-MET Y1234/1235 (#3077S, 1:1000; Cell Signaling), MET (#8198S, 1:1000; Cell Signaling), p-Akt S473 (#4060L, 1:1000; Cell Signaling), Akt (#9272S, 1:1000; Cell Signaling), p-ERK1/2 T202/Y204 (#4370S, 1:1000; Cell Signaling), ERK1/2 (#4695S, 1:1000; Cell Signaling), p-STAT3 Y705 (#4370S, 1:1000; Cell Signaling), STAT3 (#9139S, 1:1000; Cell Signaling), p-c-Jun S63 (#9261S, 1:1000; Cell Signaling), c-Jun (#9165S, 1:1000; Cell Signaling), p-ETS1 T38 (#ab59179, 1:1000; Abcam, Cambridge, UK), ETS1 (#14069S, 1:1000; Cell Signaling), β-Actin (#sc-47778, 1:2500; Santa Cruz, Dallas, TX), Histone-H3 (#4499S, 1:2000; Cell Signaling) and vinculin (#V9131, 1:1000; MilliporeSigma). Horseradish peroxidase (HRP)-conjugated secondary antibodies (#G21234 and #G21040, 1:2000; Thermo Fisher Scientific) were used according to the manufacturers' instructions. Quantitative analysis was performed by using ImageJ software (NIH, Bethesda, MD).

For EGF- or HGF-induced degradation of EGFR or MET, after siRNA transfection, the cells ($5 \times 10^5$) were seeded in each well of a 6-well plate containing RPMI 1640 with 10% FBS. Eight hours after seeding, cells were washed 3 times with PBS and starved in RPMI 1640 without FBS for 16 h. The cells then were untreated or treated with 100 ng/ml of EGF or HGF in the presence of cycloheximide (40 μg/ml) for the indicated times. After the stimulation, cells were washed 3 times with PBS and harvested/resuspended in 150–200 μl of 2× Laemmli buffer, and the cell lysates were subjected to western blotting and image analysis as described earlier.

## Endocytic recycling assay

TfnR recycling assays were performed using biotinylated Tfn, which was conjugated at a 7:1 molar ratio with the cleavable EZ-Link Sulfo-NHS-SS-Biotin (#A39258, Thermo Fisher Scientific) according to the manufacturer's instructions. For EGFR recycling, non-cleavable biotinylated EGF (#E3477; Thermo Fisher Scientific) was used for assays. TfnR and EGFR recycling assays were performed as previously described [13]. In brief, cells were grown overnight in gelatin-coated 96-well plates at a density of $3 \times 10^4$ cells/well and incubated with 10 μg/ml biotinylated Tfn or 20 ng/ml biotinylated EGF in assay buffer (PBS$^{4+}$: PBS supplemented with 1 mM MgCl$_2$, 1 mM CaCl$_2$, 5 mM glucose, and 0.2% bovine serum albumin) for either a 10- or 30-min pulse at 37 ˚C. Cells were then immediately cooled down (4 ˚C) to stop internalization. The remaining surface-bound biotinylated Tfn was cleaved by incubation with 10 mM Tris (2-carboxyethyl) phosphine (TCEP) in assay buffer for 30 min at 4 ˚C. The surface-bound biotinylated EGF was removed by acid wash (0.2 M acetic acid, 0.2 M NaCl [pH 2.5]) at 4 ˚C. For TfnR recycling assays using ERK1/2 inhibitors (SCH772984), cells were incubated in the absence or presence of 10 μM of SCH772984 in the assay buffer containing 10 mM TCEP for 30 min at 4 ˚C before recycling assays were performed in the continued absence or presence of the inhibitor. Cells were washed with cold PBS$^{4+}$ buffer and then incubated in PBS$^{4+}$ containing 2 mg/ml of holo-Tfn (#T0665, MilliporeSigma, Burlington, MA) or 100 ng/ml of EGF and 10 mM of TCEP at 37 ˚C for the indicated times. The recycled biotinylated Tfn or biotinylated EGF was removed from the cells by the acid wash step. Cells were then washed with cold PBS and then fixed in 4% paraformaldehyde (PFA) (Electron Microscopy Sciences, Hatfield, PA) in PBS for 30 min and further permeabilized with 0.1% Triton X-100/PBS for 10 min. Remaining intracellular biotinylated ligands were assessed by streptavidin-POD (#11089153001, 1:10000; Roche, Basel, Switzerland) in Q-PBS, which contains 0.2% BSA (Equitech-Bio, Kerrville, TX), 0.001% saponin (MilliporeSigma), and 0.01% glycine (MilliporeSigma). The reaction was further developed with OPD (MilliporeSigma) and then stopped by addition of 50 μl of 5 M of H$_2$SO$_4$. The absorbance was read at 490 nm (Synergy H1 Hybrid Reader, Biotek, Winooski, VT). The decrease in intracellular biotinylated ligands (recycling) was represented as the percentage of the total internal pool of ligand internalized. Well-to-well variability in cell number was accounted for by normalizing the reading at 490 nm with a BCA readout at 562 nm.

## Immunofluorescence and confocal microscopy analyses

After siRNA transfection, the cells were washed 3 times with PBS and then starved in RPMI 1640 medium without FBS for 30 min at 37 ˚C. Cells were incubated with 20 ng/ml EGF or 1 μg/ml HGF in RPMI 1640 medium for 30 min at 4 ˚C, washed with cold PBS, and then incubated in pre-warmed RPMI 1640 medium at 37 ˚C for the indicated times. Cells were then washed with ice-cold PBS to stop chase, fixed with 4% (w/v) PFA for 30 min at 37 ˚C, and permeabilized using 0.05% saponin (w/v) (MilliporeSigma) for 10 min. Cells were blocked with Q-PBS and probed with antibodies against the following proteins: p-EGFR Y1068 (#3777S,

1:200; Cell Signaling), MET (#AF276, 1:100; R&D Systems, Minneapolis, MN), EEA1 (#610457, 1:100; BD Biosciences, San Jose, CA), Rab11 (#5589S, 1:50; Cell Signaling), and LAMP1 (#ab25245, 1:75; Abcam, Cambridge, UK), according to the manufacturers' instructions. AlexaFluor-conjugated secondary antibodies (#A-11036, A-11055, A-21206, A-21434, A-31571; Thermo Fisher Scientific) were used according to the manufacturers' instructions. Fixed cells were mounted in PBS and imaged using a 60×, 1.49 NA APO objective (Nikon, Tokyo, Japan) mounted on a Nikon Ti-Eclipse inverted microscope coupled to an Andor Diskovery Spinning disk confocal/Borealis widefield illuminator equipped with an additional 1.8× tube lens (yielding a final magnification of 108×). The pinhole size was 50 μm. The percentages of colocalizations were determined using ImageJ software (NIH, Bethesda, MD).

## Quantitative reverse transcription PCR analysis

Cell samples were homogenized, and total RNA was isolated using TRIzol reagent (Thermo Fisher Scientific) and treated with DNase I (#AMPD1; MilliporeSigma) according to the manufacturer's instructions. Equal microgram volumes of RNA were reverse transcribed for each experimental condition by the use of a high-capacity cDNA reverse transfection kit (#4368813; Thermo Fisher Scientific) with oligo dT (#12577–011; Thermo Fisher Scientific). qPCR was carried out using SYBR green reagents in technical duplicates (Thermo Fisher Scientific) on an ABI QuantStudio 5 Real-Time PCR System; primer sequence information is in Table 1. At least 3 independent experiments were performed for all experiments described, and all results presented were calculated from CT values derived from the quantitative reverse transcription PCR (qRT-PCR) reactions. All values were normalized to *GAPDH*.

## Nuclear/Cytosol fractionation

After siRNA transfection, the cells were subjected to fractionation using the Nuclear/Cytosol Fractionation Kit (#K266-25; BioVision, Milpitas, CA) according to the manufacturer's instructions.

## Rab7 activation assay

After siRNA transfection, the cells were collected in ice-cold PBS containing protease and phosphatase inhibitor cocktails (Roche, Basel, Switzerland). The cells were used to measure

**Table 1.** **Primer sequences.**

| Target gene | Primer sequences |
| --- | --- |
| Human EGFR F | 5'-CAAGGAAGCCAAGCCAAATG-3' |
| Human EGFR R | 5'-CCGTGGTCATGCTCCAATAA-3' |
| Human MET F | 5'-CCACGGGACAACACAATACA-3' |
| Human MET R | 5'-TAAAGTGCCACCAGCCATAG-3' |
| Human c-Jun F | 5'-AAGAACTCGGACCTCCTCA-3' |
| Human c-Jun R | 5'-CCGTTGCTGGACTGGATTAT-3' |
| Human HGF F | 5'-GGTAAAGGACGCAGCTACAA-3' |
| Human HGF R | 5'-AGCTGTGTTCGTGTGGTATC-3' |
| Human c-Fos F | 5'-CTTCCTGTTCCCAGCATCAT-3' |
| Human c-Fos R | 5'-CTGCATAGAAGGACCCAGATAG-3' |
| GAPDH F | 5'-GATTCCACCCATGGCAAATTC-3' |
| GAPDH R | 5'-CTGGAAGATGGTGATGGGATT-3' |

**Abbreviations**: c-Fos, proto-oncogene c-Fos; c-Jun, proto-oncogene c-Jun; EGFR, epidermal growth factor receptor; F, forward; GAPDH, glyceraldehyde 3-phosphate dehydrogenase; HGF, hepatocyte growth factor; MET, proto-oncogene c-Met; R, reverse

Rab7 activation using the Rab7 Activation Assay Kit (#NEBB40025; NewEast Biosciences, King of Prussia, PA) according to the manufacturer's instructions.

### Analysis of Kaplan-Meier survival data

NSCLC patient survival data were downloaded from the Kaplan-Meier plotter database [44]. Analysis of NSCLC patients was performed in FCHSD2 or Rab7 high- and low-expression cohort. *P* value was calculated by logrank test [44].

### Immunohistochemistry and image analyses

The human lung adenocarcinoma tissues were from US Biomax, Rockville, MD (#LC641) and the UT Southwestern Tissue Resource, a shared resource at the Simmons Comprehensive Cancer Center, Dallas, TX. The tumors were classified according to the American Joint Committee on Cancer (AJCC) TNM system. The immunohistochemical staining of FCHSD2 (#PA5-58432; Thermo Fisher Scientific) was optimized and performed by the core facility. The immunohistochemical images were analyzed using the IHC Profiler, ImageJ software (NIH) as previously described [45] to classify the intensities of staining. The immunoreactivity was determined by H-score, generated by adding the percentage of strong staining (3×), the percentage of moderate staining (2×), and the percentage of weak staining (1×) samples [46].

### Statistical analysis

GraphPad Prism (GraphPad Software, San Diego, CA) was used to perform statistical analyses. Data were assessed by two-tailed, unpaired Student *t* tests as indicated. $P < 0.05$ was considered significant.

### Supporting information

**S1 Data. Underlying numerical data and statistical analysis for figure panels Figs 1B, 1C, 1E, 1G, 2A, 2C, 3A, 3C, 3E, 4B, 4C, 4E, 4G, 5A, 5B, 5C and 6B; S1, S2B and S2D Figs.** (XLSX)

**S1 Raw Images. Original images supporting all blot results reported in Figs 1A, 1F, 2B, 3B, 3D, 4A, 4D, 4F, 5B and 5C; S2A, S5, S6 and S7 Figs.** The loading order, experimental samples, and molecular weight markers are indicated. The lanes used in the final figure are marked with a red box.
(PDF)

**S1 Fig. The effects of FCHSD2 depletion and ERK1/2 inhibition in TfnR endocytic recycling.** Endocytic recycling of TfnR was measured in control or FCHSD2 siRNA-treated HCC4017 and ARPE-19 cells in the absence or presence of the ERK1/2 inhibitor SCH772984 (10 μM). Cells were pulsed for 30 min with 10 μg/ml biotinylated Tfn, stripped, and reincubated at 37 ˚C for the indicated times before measuring the remaining intracellular Tfn. Percentage of recycled biotinylated Tfn was calculated relative to the initial loading. Data represent mean ± SEM ($n = 3$). Two-tailed Student *t* tests were used to assess statistical significance versus siCtrl. $^{*}P < 0.05$, $^{**}P < 0.005$. The underlying data for this figure can be found in S1 Data. ERK1/2, extracellular signal-regulated kinase 1 and 2; FCHSD2, FCH/F-BAR and Double SH3 Domain-Containing Protein; siCtrl, control siRNA; siRNA, small interfering RNA; Tfn, transferrin; TfnR, transferrin receptor.
(TIF)

**S2 Fig. FCHSD2 regulates EGFR endocytic trafficking in H1975 cells.** (A) The KD of FCHSD2 in control or FCHSD2 siRNA-treated H1975 cells. (B) Endocytic recycling of EGFR was measured in control or FCHSD2 siRNA-treated H1975 cells. Cells were pulsed for 10 min or 30 min with 20 ng/ml biotinylated EGF, stripped, and re-incubated at 37 ˚C for the indicated times before measuring the remaining intracellular EGF. Percentage of recycled EGF was calculated relative to the initial loading. Data represent mean ± SEM ($n$ = 3). Two-tailed Student $t$ tests were used to assess statistical significance. $^{*}P < 0.05$, $^{**}P < 0.005$, $^{***}P < 0.0005$. (C) Representative confocal images of pEGFR and LAMP1 immunofluorescence staining in control or FCHSD2 siRNA-treated H1975 cells. Cells were incubated with 20 ng/ml EGF for 30 min at 4 ˚C, washed, and re-incubated at 37 ˚C for the indicated times. Scale bar, 12.5 μm. (D) Colocalization of pEGFR and LAMP1 immunofluorescence staining in the cells as described in (C). Data were obtained from at least 40 cells in total/condition and represent mean ± SEM. Two-tailed Student $t$ tests were used to assess statistical significance. $^{*}P < 0.05$. The underlying data for this figure can be found in S1 Data. EGF, epidermal growth factor; EGFR, epidermal growth factor receptor; FCHSD2, FCH/F-BAR and Double SH3 Domain-Containing Protein; KD, knockdown; LAMP1, lysosome-associated membrane glycoprotein 1; siCtrl, control siRNA; siRNA, small interfering RNA.
(TIF)

**S3 Fig. Representative confocal images of MET, EEA1, and Rab11 immunofluorescence staining in control or FCHSD2 siRNA-treated HCC4017 cells.** Cells were incubated with 1 μg/ml HGF for 30 min at 4 ˚C, washed, and re-incubated at 37 ˚C for the indicated times. Scale bar, 25 μm. Quantified results are shown in Fig 2A. EEA1, early endosome antigen 1; FCHSD2, FCH/F-BAR and Double SH3 Domain-Containing Protein; HGF, hepatocyte growth factor; MET, proto-oncogene c-Met; Rab11, Ras-related protein Rab-11A; siCtrl, control siRNA; siRNA, small interfering RNA.
(TIF)

**S4 Fig. Representative confocal images of MET, EEA1, and LAMP1 immunofluorescence staining in control or FCHSD2 siRNA-treated HCC4017 cells.** Cells were incubated with 1 μg/ml HGF for 30 min at 4 ˚C, washed, and re-incubated at 37 ˚C for the indicated times. Scale bar, 25 μm. Quantified results are shown in Fig 2A. EEA1, early endosome antigen 1; FCHSD2, FCH/F-BAR and Double SH3 Domain-Containing Protein; LAMP1, lysosome-associated membrane glycoprotein 1; MET, proto-oncogene c-Met; siCtrl, control siRNA; siRNA, small interfering RNA.
(TIF)

**S5 Fig. FCHSD2 depletion-induced up-regulation of the RTKs is independent of their activities.** H1975 control or FCHSD2 siRNA-treated cells were incubated with EGFR inhibitor (afatinib) or MET inhibitor (crizotinib) as indicated concentration for 24 h. EGFR, epidermal growth factor receptor; FCHSD2, FCH/F-BAR and Double SH3 Domain-Containing Protein; MET, proto-oncogene c-Met; RTK, receptor tyrosine kinase; siCtrl, control siRNA; siRNA, small interfering RNA.
(TIF)

**S6 Fig. FCHSD2 deficiency increases MET expression and ERK1/2 activity.** The HCC4017 cells stably expressing shCtrl or shFCHSD2 at steady state. ERK1/2, extracellular signal-regulated kinase 1 and 2; FCHSD2, FCH/F-BAR and Double SH3 Domain-Containing Protein; MET, proto-oncogene c-Met; shCtrl, control shRNA; shRNA, small hairpin RNA.
(TIF)

**S7 Fig. Rab7 is essential for FCHSD2 deficiency-induced up-regulation of MET.** Rab7 KD by a different pool of siRNAs abolishes the MET up-regulation induced by FCHSD2 depletion. FCHSD2, FCH/F-BAR and Double SH3 Domain-Containing Protein; KD, knockdown; MET, proto-oncogene c-Met; Rab7, Ras-related protein Rab-7A; siCtrl, control siRNA; siRNA, small interfering RNA.
(TIF)

## Acknowledgments

We thank members of the Schmid lab for critically reading the manuscript, especially Marcel Mettlen for help in preparing illustrations and Kim Reed for technical assistance in plasmid preparation. We thank colleagues from the Department of Biophysics: Drs. Emiko Uchikawa and Xiaochen Bai for kindly providing recombinant human HGF. We acknowledge the UT Southwestern Tissue Resource, a shared resource at the Simmons Comprehensive Cancer Center, for assistance with immunohistochemistry data analysis and interpretation.

## Author Contributions

**Conceptualization:** Guan-Yu Xiao, Sandra L. Schmid.

**Data curation:** Guan-Yu Xiao.

**Formal analysis:** Guan-Yu Xiao.

**Funding acquisition:** Sandra L. Schmid.

**Investigation:** Guan-Yu Xiao.

**Methodology:** Guan-Yu Xiao.

**Supervision:** Sandra L. Schmid.

**Validation:** Guan-Yu Xiao.

**Writing – original draft:** Guan-Yu Xiao, Sandra L. Schmid.

**Writing – review & editing:** Guan-Yu Xiao, Sandra L. Schmid.

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
