## [Editor Report · Decision Letter 0]

12 Feb 2020

Dear Sandy, 

Thank you for submitting your manuscript entitled "FCHSD2 Controls Oncogenic ERK1/2 Signaling Outcome by Regulating Endocytic Trafficking" for consideration as a Research Article by PLOS Biology.

Your manuscript has now been evaluated by the PLOS Biology editorial staff as well as by an academic editor with relevant expertise and I am writing to let you know that we would like to send your submission out for external peer review.

Please re-submit your manuscript within two working days, i.e. by Feb 14 2020 11:59PM.

Kind regards,

Ines

--

Ines Alvarez-Garcia, PhD

Senior Editor

PLOS Biology

Carlyle House, Carlyle Road

Cambridge, CB4 3DN

+44 1223–442810

---

## [Decision Letter · Decision Letter 1]

11 Mar 2020

Dear Sandy,

Thank you very much for submitting your manuscript "FCHSD2 Controls Oncogenic ERK1/2 Signaling Outcome by Regulating Endocytic Trafficking" for consideration as a Research Article at PLOS Biology. Your manuscript has been evaluated by the PLOS Biology editors, an Academic Editor with relevant expertise, and by three independent reviewers.

As you will see, all the reviewers are very positive and find the conclusions interesting and significant for the field, however they also raise a few points that would need to be addressed with additional experiments to strengthen the results. The reviewers have also requested several clarifications.

In light of the reviews (attached below), we will not be able to accept the current version of the manuscript, but we would welcome re-submission of a revised version that takes into account the reviewers' comments. We cannot make any decision about publication until we have seen the revised manuscript and your response to the reviewers' comments. Your revised manuscript is also likely to be sent for further evaluation by the reviewers.

We expect to receive your revised manuscript within 2 months. 

**IMPORTANT - SUBMITTING YOUR REVISION**

*Re-submission Checklist*

*Published Peer Review*

*PLOS Data Policy*

*Blot and Gel Data Policy*

Best wishes,

Ines

--

Ines Alvarez-Garcia, PhD

Senior Editor

PLOS Biology

Carlyle House, Carlyle Road

Cambridge, CB4 3DN

+44 1223–442810

Reviewers’ comments

Rev. 1:

In the present manuscript, Xiao G. et al., characterized the role of FCHSD2 in endocytosis and trafficking of RTKs. Authors reported that FCHSD2 has a role in ligand-induced trafficking of EGFR and Met receptors, and, in particular, they proposed that it acts as a positive regulator of receptor recycling. They also characterized the role of FCHSD2 in unstimulated basal conditions, where depletion of FCHSD2 causes the increased transcription of EGFR and MET mRNA, possibly through the activation and nuclear translocation of ERK1/2/c-Jun. They also showed that Rab7 is required for the decreased recycling of TfR and EGFR upon siFCHSD2, suggesting that FCHSD2 function could be that of counteracting Rab7. In addition, siRab7 also reverted the increase in EGFR and MET levels induced upon FCHSD2 depletion. FCHSD2 loss correlates with high grade NSCLC; patients with high FCHSD2 have an elevated survival at variance with patients with high Rab7 that display low survival rates.

The manuscript provides novel insight into the connection between endocytosis and signaling in cancer. Therefore, is of high interest for the cell biology and cancer community. However, there are a number of issues that need to be solved prior publication.

1) The assay described in Fig. 1B, C does not measure properly recycling, but it just detects disappearance of internalized Tf/EGF signal. For TfR, which is almost exclusively recycled back to the PM, it could be used as a proxy for recycling. However, this is not the case for EGFR, which is both recycled and degraded after internalization. Thus, the experiment in Fig. 1C suggests that disappearance of internalized EGF signal is delayed, which is compatible with either defective recycling or defective degradation. This methodological issue need to be better discussed. In addition, the nomenclature on the Y axis in Fig. 1B, C need to be changed accordingly. As it is represented now, it is misleading as it seems that recycling decreases with time.

2) Fig. 1F and Fig. 2B. It is not clear how the assay is performed. Are all samples (from 0.5h to 6h of EGF stimulation) treated with cycloheximide for the same period of time? As cycloheximide could really affect the biosynthesis of many factors involved in endocytosis, trafficking and degradation (also considering the effects of FCHSD2 on transcription/biosynthesis), I would recommend to repeat EGFR degradation curve (-/+ siFCHSD2) without this treatment.

3) Fig. 3B. The treatment with Afatinib is not convincing, as it is not substantially reducing EGFR activation (measure as p-EGFR). More consistent inhibition is required to exclude a role for EGFR kinase activity in the process. Also, the WB shown on the left does not reproduce the increase in EGFR and MET upon siFCHSD2 observed in previous experiments. As Erk1/2 is involved in the process, another standard loading control should be used.

4) Fig. 3D. Authors concluded from this experiment that STAT3 is not involved in the process. However, its phosphorylation is affected upon siFCHSD2. Authors should better discuss their conclusion.

5) Fig. S1. The inhibition of Erk1/2 with SCH compound has the same effect of the Rab7 inhibition on TfR recycling (in Fig. 5A). It seems that the decrease in TfR recycling upon siFCHSD2 requires the action of Erk1/2, as it is reverted by SCH treatment. I did not understand how the authors interpret this results in Fig. S1 at variance with results shown in Fig 5 A.

In addition, in Fig. S1, the effects of SCH treatment are mild - if any - as well as the effects of siFCHSD2 in ARPE-19 cells. The magnitude is minimal and, although some time points are statistically significant, it is hard to draw conclusion on the biological significance of these effects. In my opinion, these data on SCH treatment should be more critically discussed.

6) Fig. 4F. The WB provided is not completely convincing. The levels are variable and there are some inconsistencies. For instance, MET levels seem not to be rescued by both treatments in H1975, just all levels are decreased. Similarly, in HCC4079, MET levels are variable, sometimes increasing (SCH) sometimes decreasing (GSK) after treatments. It is not clear what these treatments are really doing. In addition, the blots are not representative of the quantification on the right. In the quantitation, samples should be all normalized to the same reference, i.e. the not treated Ctrl sample.

An important experiment that would corroborate this finding is the analysis of EGFR and MET mRNA levels upon siFCHSD2, to investigate if they are rescued by these treatments.

7) The effect of siRab7 are strong and convincing. However, it is not clear how Rab7 function is linked to Erk1/2 nuclear translocation and activation, and to the induced EGFR/MET transcription. The model shown in Fig. 6A is too complicated and does not convey the message.

Which is the mechanism though which Rab7 KD revert the increased EGFR and MET levels? Authors should show if siRab7 is able to revert the MET and EGFR mRNA levels.

Does siRab7 revert the increased Erk1/2, c-Jun, Ets-1 levels in the nucleus?

These connections need to be investigated and further discussed in the text.

Rev. 2:

This is an excellent study interrogating how membrane trafficking aletrations downstream of FCHSD2 depletion impact ERK1/2 signaling. The primary conclusions are that FCHSD2, by negative regulation of Rab7, diverts tyrosine kinase receptors (TKRs) from late endosomes, limiting access of TKRs to ERK1/2 activation at the late endosome. Loss of FCHSD2 increases traffic of TKRs to LE and subsequent activation of ERK1/2. The data, for the most part, support the stated conclusions. My main comments are as follows.

1. The authors should be more attentive to alerting the reader what was done in the KRAS/p53 line and what was done in EGFR mutant. For the non-cancer specialist, the potential importance of the distinction might be lost. The authors might also consider adding a discussion paragraph on the topic of the oncogenic driver mutation and the impact of FCHSD2 (loss thereof).

2. The authors might consider extending the main conclusion, that KD of FCHSD2 increases expression of EGFR and MET, in analyses of other human NSCLC cell lines, both EGFR mutant and KRAS.

3. The authors might provide some speculation (or perhaps data) on how FCHDS2 acts as a negative regulator of Rab7. Does FCHSD2 activate a Ran7 GAP or inhibit a Rab7 GEF?

4. Fig 6 is not particularly strong, and I do not feel it strengthens the paper but rather detracts from the otherwise strong cell biology. To appreciate the data in panel B one would want to see expression of FCHSD2 in adjacent normal lung tissue. Pushing a link between the expression of a single protein (FCHSD2) and grade is not particularly compelling. I assume the analyses in Fig6C are based on RNAseq. Have they correlated IHC with RNAseq in their own cohort (panel B) to establish the RNAseq data agrees with the IHC? The author's need to present data on the expression distribution of FCHSD2 and Rab7 so the reader can assess how meaningful the distinction is. Did the author's use above and below the median or upper and lower quartiles? They should also look at the TCGA database to determine how specific this is for lung.

5. Throughout the manuscript the authors should reanalyze the recycling curves. In many instances it looks like the extent of recycling rather than the rate of recycling is changed. For example, in Fig 1B it is the extent of transferrin receptor recycling that is changed by FCHSD2 depletion not the rate, suggesting a fraction of transferrin receptor is trapped and no longer recycled rather than a slowing of recycling. Although I do not think it will change the conclusions it might change how one thinks about the molecular mechanism(s) underlying the effects.

Rev. 3: Peter S McPherson – Please note that this reviewer has waived anonymity

This is a clear, concise, well-executed study, exactly what we have come to expect from Dr. Schmid and her colleagues. I was left a little wanting, how exactly does the FCHSD2-depeletion induced upregulation of the RTKs occur, what specifically does Rab7 do in this pathway, why the seeming conundrum between increased trafficking to late endosome/lysosome and yet upregulation of expression? But there is valuable information in this study and no one study ever has all the answers.

Abstract. "…can reciprocally regulate endocytic trafficking by creating feedback loops in non-small-cell lung cancer (NSCLC) cells…". The sentence is awkward. Presumably these feedback loops are present in other cancer cells, perhaps in non-cancer cells.

Line 53, can you please be more specific as to what you mean by "…cancer-specific activation…", in particular, what is meant by activation?

Figure 1 and throughout, use 2 siRNAs or better describe the specificity of the single one used or perhaps validate 1-2 key phenotypes in a knockout line.

Figure 1D, please, a word or two about the specificity of the PO4-EGFR antibody in the results section.

The upregulation of mRNA in Figure 3 is relatively minor, and in light of the enhanced degradation its seems the balance should not be towards such a strong upregulation of the protein. Please discuss.

Figure 5, one Rab7 siRNA?

Figure 5, higher levels of active Rab7 in FCHSD2 knockdown cells. It would be nice if there was some additional data here. What explains the activation, regulation of a GEF or a GAP?

Figure 5. There is some uncertainty regarding the specificity of the antibodies recognizing the active Rabs. The results state that an antibody specific to active Rab7 was used without reference or validation. The authors need to address this issue.

---

## [Editor Report · Decision Letter 2]

22 May 2020

Dear Sandy,

Thank you for submitting your revised Research Article entitled "FCHSD2 Controls Oncogenic ERK1/2 Signaling Outcome by Regulating Endocytic Trafficking" for publication in PLOS Biology. I have now obtained advice from the original Academic Editor, who has assessed the revision along with the team of editors.

We're delighted to let you know that we're now editorially satisfied with your manuscript. However before we can formally accept your paper and consider it "in press", we also need to ensure that your article conforms to our guidelines. A member of our team will be in touch shortly with a set of requests. As we can't proceed until these requirements are met, your swift response will help prevent delays to publication. Please also make sure to address the data and other policy-related requests noted at the end of this email.

*Copyediting*

*Published Peer Review History*

*Early Version*

*Submitting Your Revision*

Best wishes,

Ines

--

Ines Alvarez-Garcia, PhD

Senior Editor

PLOS Biology

Carlyle House, Carlyle Road

Cambridge, CB4 3DN

+44 1223–442810

DATA POLICY:

Thank you for submitting all the data underlying the graphs shown in the figures. Please indicate in each figure legend (including the supplementary figures) where the underlying data can be found.

---

## [Editor Report · Decision Letter 3]

30 Jun 2020

Dear Dr Schmid,

On behalf of my colleagues and the Academic Editor, Frederick M. Hughson, I am pleased to inform you that we will be delighted to publish your Research Article in PLOS Biology. 

Early Version

PRESS 

Kind regards,

Vita Usova

Publication Assistant, 

PLOS Biology

on behalf of

Ines Alvarez-Garcia,

Senior Editor

PLOS Biology